# Estimation of the Energy Consumption of the Rice and Corn Drying Process in the Equatorial Zone

**Emérita Delgado-Plaza** [1,*] **, Miguel Quilambaqui** [1] **, Juan Peralta-Jaramillo** [1] **, Hector Apolo** [1] **and Borja Velázquez-Martí** [2]

1   Escuela Superior Politécnica del Litoral, ESPOL, FIMCP-CDTS, ESPOL Polytechnic University, Campus Gustavo Galindo Km, 30.5 Vía Perimetral, Guayaquil P.O. Box 09-01-5863, Ecuador; mquilamb@espol.edu.ec (M.Q.); jperal@espol.edu.ec (J.P.-J.); hapolo@espol.edu.ec (H.A.)
2   Departamento de Ingeniería Rural y Agroalimentaria, Universitat Politècnica de València, Camino de Vera s/n, 46022 Valencia, Spain; borvemar@dmta.upv.es
*   Correspondence: eadelgad@espol.edu.ec

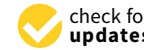

**Featured Application: This document is aimed at the agricultural sector to assess the energy consumption and problems associated with drying equipment used to dry rice and corn.**

**Abstract:** Drying is considered one of the industrial processes that requires more energy than other processes, being a topic of much interest to the agricultural sector, especially the evaluation of energy consumption for rice and corn dryers. To meet this goal, an overview survey matrix and protocols for temperature measurements of dryers were developed. The study evaluated 49 rice dryers and 14 yellow corn dryers. As a result, it was determined that the oversizing of the fan/extractor and the dryer engine generates a high energy consumption, added to the lack of insulation in the heat ducts. Therefore, the drying productivity index is very low in dryers using liquefied petroleum gas (LPG) being 0.14 dollar/quintal for rice and 0.27 dollar/quintal for corn and using biomass reaches 1.4 dollars/quintal. In relation to energy losses, these account for more than 55%. Inadequate energy management in drying processes directly influences the marketing chain of products, the losses of which are caused by fluctuations in the price of rice and corn on the domestic market, with the agricultural sector having to generate an energy efficiency plan.

**Keywords:** dryers; rice; corn; energy valuation

---

## 1. Introduction

Drying is one of the processes where the highest percentage of energy is consumed at the agroindustrial level [1]. The types and methods of drying used have significant effects on the quality of the final product. In practice, the energy requirement for drying food grains has two components; firstly, the energy required to evaporate the water vapor and secondly, the energy required to remove the water associated with the raw material [2,3].

According to Food and Agriculture Organization, there are different ways to define the drying process, considering the theoretical aspects of the energy and mass transfer mechanisms [4]. On this regard, we can define it as a natural or artificial process, which have an asynchronous exchange of heat and mass between the air existing in the drying process and the grains. In other words, we can define drying as the process capable of reducing the moisture content of a product to a minimum and safe level so that it can be consumed and stored. In this way, the respiratory activity of the grain is reduced, avoiding the insects, fungi, and molds to decrease the grain quality, produce toxins, and prevent its commercialization and consumption [2,3,5].

For the drying process, it is necessary to determine the energy consumption, therefore it is estimated the energy required to dry grains, depending on the drying temperature, ranges between 3000 to 8000 kJ/kg of water removed [5–7]. That would mean an approximate energy consumption of 0.8 to 2.2 kWh for each kilogram of dry product.

On the other hand, there is a relationship between the product to be dried and the energetic consumption required in the process. Therefore, it is essential to analyse the energy requirements to minimize the energy losses as well as use the energy input efficiently. Likewise, it is necessary to take into consideration technological variables associated with the drying system, as well as, the grain quality and initial humidity, which can considerably affect the energy demand of the process [3,5–8].

In our particular case, the Republic of Ecuador is located in the northwest of South America, and the Pacific Ocean bathes its coasts. It has 277 thousand km$^2$ and a population of 17 million inhabitants [9], where the topography plays an essential role in the distribution of the country's climates. The Andean Mountain Range, which is the leading mountain chain of South America, crosses our continental territory from North to South, dividing it into three natural regions with very different climatological characteristics. These regions are called: Coast, Highland, and Amazon. The mountain range affects the formation of cold air masses that modify the precipitation patterns and acts as a wall that prevent air masses from the Pacific Ocean and the Amazon to exchange. Moreover, the "El Niño Current" plays an important role in the Andean region climate [10].

In summary, all these factors affect local meteorological variables' behaviour, thus contributing to the presence of areas with very particular microclimates throughout the continental territory, which leads to crops sectorization and production areas with different degrees of agroindustrial development [11,12].

The Ecuadorian agricultural production was about 4524.00 billion US dollars in 2013 and contributed with 13.8% of the total GDP. It is worth mentioning that the crops that guarantee food sovereignty (rice, corn, potatoes, vegetables, fruits, milk, meat, among others) are not considered in the GDP analysis [13]. Now, concerning per capita consumption of food products prepared within the Food Balance 2000–2012 by MAGAP, it is cited that the products with the highest national demand are: corn and rice with an annual average increase of 47 and 9 kg/(person per year) respectively [14]. On the other hand, it is important to mention that 70% of corn production comes from family farming and it is a crucial sector to guarantee food security and poverty eradication [15]. From the previous evidence, it is clear that corn and rice are vital products for Ecuador's food sovereignty; therefore, it is necessary to analyse the entire production chain, including energy use.

In Ecuador, drying processes at the agroindustrial level mainly use liquefied petroleum gas (LPG) and diesel fuel as an energy source. However, the cost of both energy sources depends on the international market. Locally, the policy of subsidies about fuels makes LPG for domestic use the most widely used in the drying industry, since its price is 0.107 $/kg [16]; it is lower compared to the LPG of agricultural use which price is 0.188 $/kg or diesel with a price of 1781$/gallon [17]. The situation described, concerning costs, stimulate the consumption of LPG for domestic use in most agroindustrial processes, which directly affects economic resources and targeting of subsidies.

This problem forced the Ecuadorian State to issue Ministerial Agreement No. 69 in 2007, which aimed to regularize the use of LPG by users of the agroindustrial sector dedicated to grain drying (corn, rice, and soybean), establishing the obligation to install and use centralized facilities or systems for receiving LPG. To date, the regulation of LPG use has not yet been achieved, mainly due to technical problems and ignorance of the energy consumption during the drying processes.

From the energy point of view, another variable of interest is electricity consumption, which, in the Ecuadorian case, is regulated using a tariff sheet issued by the Electricity Regulation and Control Agency, classifying the consumer as residential, commercial, and industrial. In the particular case of residential consumers, the application of the Dignity Rate Subsidy is provided when monthly energy consumption is less than 110 kWh-month in the distribution companies of the Highland Region and 130 kWh-month in those of the Coast/Amazon/Galapagos Region [18], with an average cost of

4 ¢/kWh, [19] compared to the invoiced cost of the residential consumer, which is 10.28 ¢/kWh. On the other hand, consumers of the industrial category have a billing cost that depends on its consumption range and the rate of the voltage level they use; in practice, it is an average value of 7.28 ¢/kWh. In summary, in Ecuador, the cost of electricity service for the year 2019 reached an average price of 9 ¢/kWh [20]. It is appropriate to emphasize that, in the Ecuadorian case, in addition to explicit consumption subsidies, there are subsidies to the implicit electricity derived from the hidden costs of electricity, related, for example, to the fact that public generation companies exclude capital costs from calculating the cost of energy, transmission, and distribution [21].

In the particular case of Ecuador, there is little information on energy consumption in drying processes. In addition, there is no assessment between the existing drying technology and its association with the designs and construction of dryers in agroindustrial plants or collection centres of cereals on the Ecuadorian coast [22]. Moreover, most dryers in the agroindustrial sector are located in the provinces of Guayas and Los Ríos, which corresponds to the climatic floor of "Tropical Mega thermic semi-humid," according to the classification established by Pourrut in 1983 [12]. Most equipment is of the artisanal or semiartisan type; therefore, there are many problems during drying due to not being able to control the temperature and wind speed of the hot gas towards the drying chamber (nonuniform drying). In addition, due to its poorly designed and constructed equipment consumes an unnecessary amount of fuel for its operation [23].

The majority of the rice collection centres (piling centres) are located in the cantons of Daule, Salitre, and Santa Lucía, with more than 200 rice hullers. Regarding to corn, its collection centres are located in Los Ríos province, especially in Ventanas canton, which is considered a corn city due to its agricultural nature [24].

### 1.1. Rice and Corn Production

The work starts from the analysis of a representative sample number of rice and corn dryers to assess the energy consumption used in the equatorial zone. Regarding that, a bibliographic compilation of the statistical data published by the National Institute of Statistics and Census (INEC) of the year 2018 was used. Mentioned INEC research, shows Ecuador's provinces with the highest rice and yellow corn production. As can be seen in Figure 1, Guayas province is the largest rice producer. Regarding the cultivation and production of corn, Los Ríos province is the largest producer.

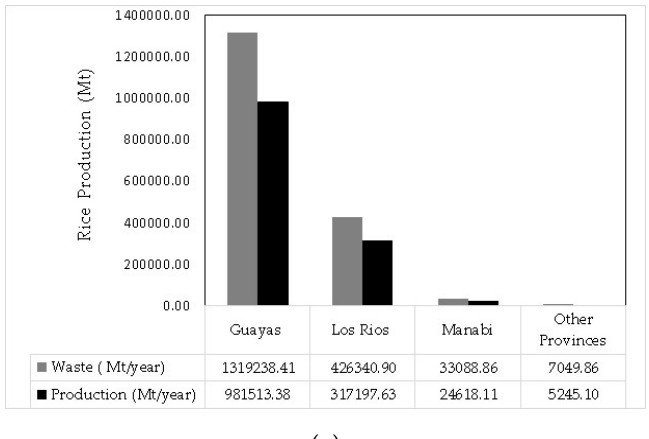

(a)

**Figure 1.** *Cont.*

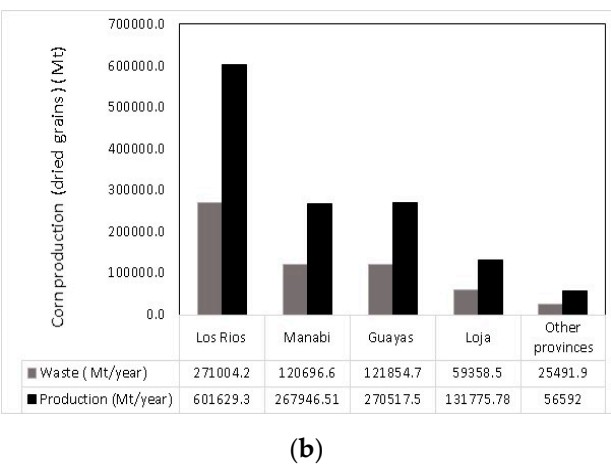

(**b**)

**Figure 1.** Production metric tons (Mt) (**a**) rice, (**b**) corn. INEC, 2018.

*1.2. Rice Processing Plant*

From the section previously analysed, we proceed to investigate the number of rice processing plants (called "Piladoras") located in the province of Guayas and its area of operation and the yellow corn collection centres located in the Los Ríos province. The information from the census was carried out by the Ministry of Agriculture and Livestock (MAGAP) in 2018. It is essential to clarify that rice processing plants and corn collection centres have dryers as the first step of the production process.

The MAGAP census establishes 581 rice processing plants in the Guayas province, categorized among large, medium, and small. This categorization is granted based on the production of quintals/hour(h) of drying; for a greater understanding in this work, it is express in kg/h. In the case of large rice processing plants (considered as the first category), their production would be around 2045–11,000 kg/h, medium (second category) between 1136 and 1800 kg/h, and small (third category) between 227 to 900 kg/h [24–26]. Figure 2a shows statistical data on the number of rice processing plants by production size. As can be seen, 65% correspond to small production plants, indicating that the technology used in this sector, especially the dryers, is handcrafted. This information visualizes in Figure 2b, where the classification of rice processing plants by type of technology is presented, showing that 58% of the plants use artisan-built equipment.

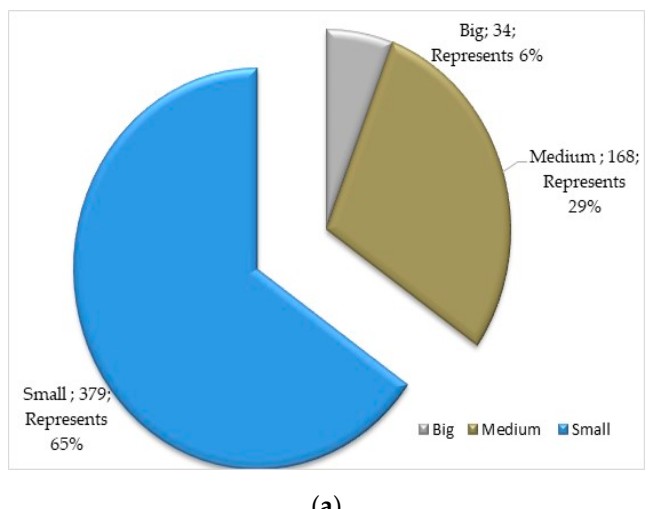

(**a**)

**Figure 2.** *Cont*.

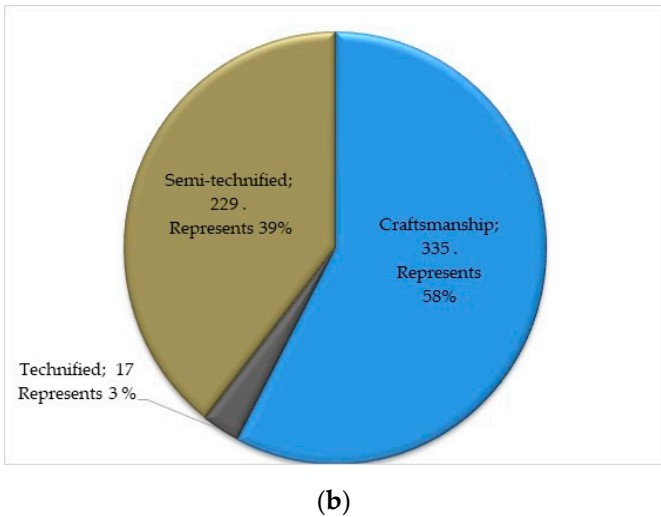

(**b**)

**Figure 2.** (**a**) Classification of rice processing plants by production size, (**b**) type of technology. Data from MAGAP, 2018.

In order to reach a greater reduction of the sample, we review the information on the areas with the highest concentration of rice processing plants in the province of Guayas. Locating those in Daule, Salitre, and Santa Lucia cantons, corresponding to 90, 87, and 54 rice processing plants respectively (Figure 3). With a population size of 231 rice processing plants, the sample is determined from this using the sample size formula through the respective statistical method, with a confidence level of 95% and a margin of error of 5%; therefore, the sample reduces to 68 rice processing plants.

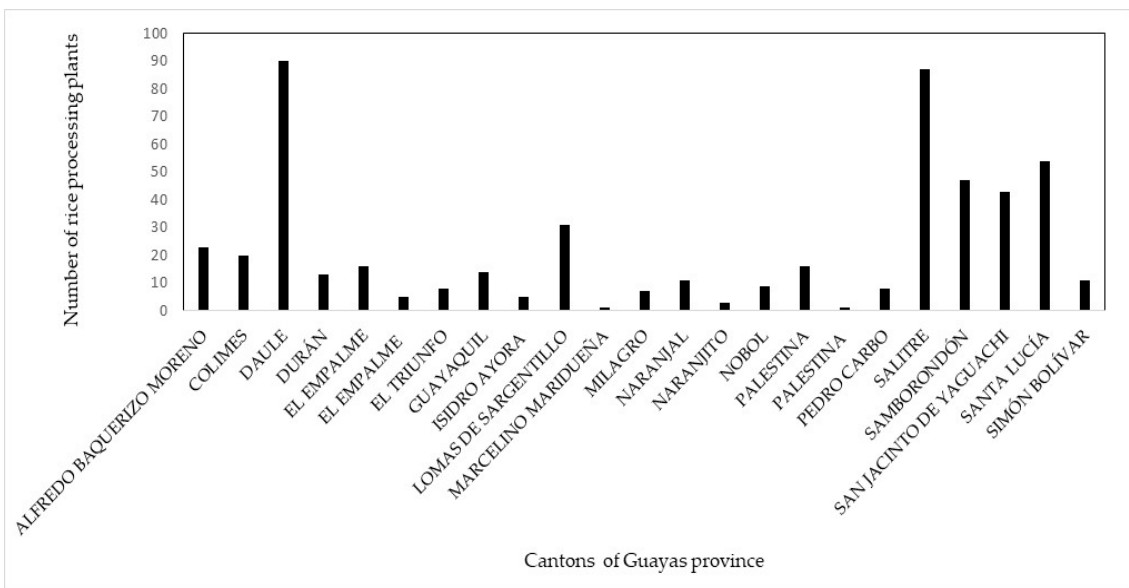

**Figure 3.** Rice processing plants in the cantons of Guayas province. Fuente. Geoportal MAGAP, 2018.

Continuing with the analysis, we review information registered with the Ministry of Agriculture and Livestock (MAGAP) about the rice processing plants which have drying equipment. According to that information, small rice processing plants, considered as a third category, use "tendales" (open space to spread food to be dried naturally "sun-dried") for drying the rice.

Figure 4 shows the number of rice processing plants per category that use these two drying processes: natural or artificial (dryers). For this research we will evaluate the second and third categories of rice processing plants energy consumption, since they mostly have artisan-built equipment.

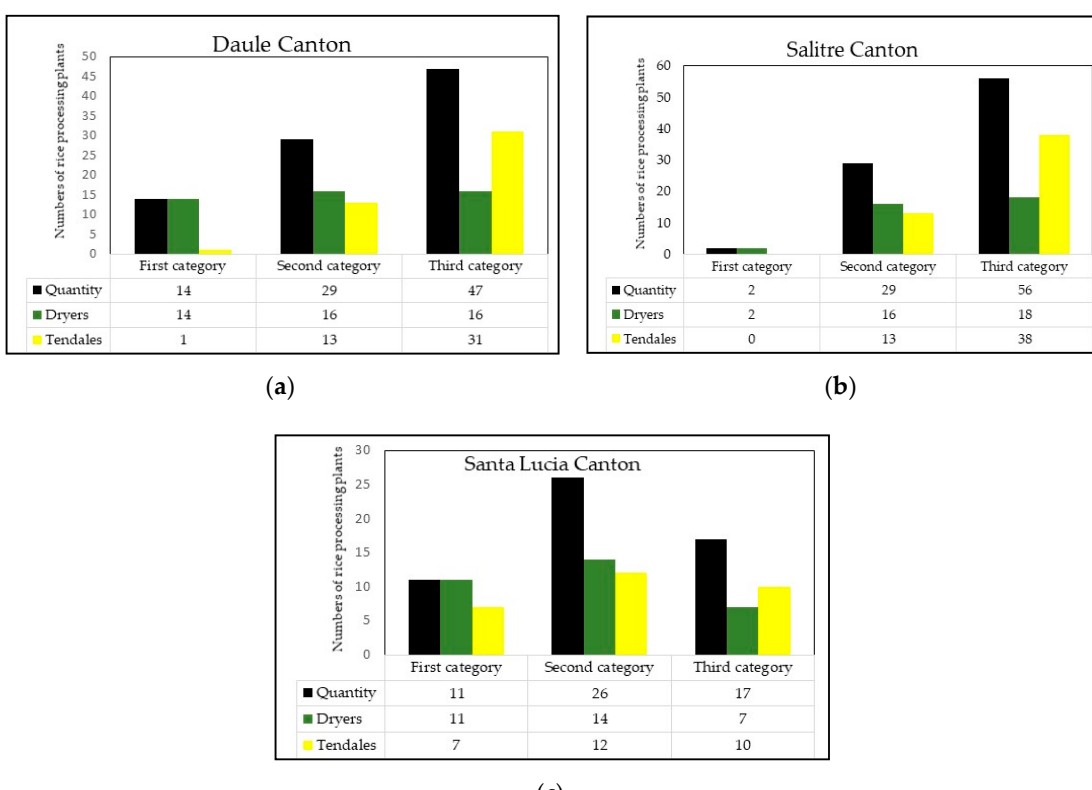

**Figure 4.** Number of rice processing plants per category per canton; (**a**) Daule, (**b**) Salitre, and (**c**) Santa Lucia. MAGAP, 2018.

## 1.3. Corn Processing Plants

In corn, the highest concentration of collection centres is in the province of Los Ríos in cantons such as Quevedo and Ventanas. They are classified in the corn sector, such as: A and A ++ (size of the company). On the other hand, small collection centres can process up to 40 quintals/hour of dry corn, for medium and large collection centres they produce between 40 and 60 quintals/hour and 60 and 75 quintals/hour, respectively (information taken off the corn centres visited).

## 2. Methods

### Technological Evaluation of the Dryers

In assessing the drying technology, the investigative method was used to prepare technical sheets to gather information about the technology to be studied. These tools elaborate from the dependent variables necessary to evaluate the equipment's energy consumption and the parameters established in food products drying kinetics.

Before preparing the technical sheet, it is necessary to know the main parts of a dryer: drying chamber, exhaust fan, and burner (Figure 5) [27,28].

From the bibliographic review of several authors, a technical file was prepared which consists of five essential parts [6,29]:

(a) *General information of the company and the producer*: company name, address, coordinates, MAGAP category, monthly production, type of product, product origin, % impurity and technology used (National, International, empirical construction).

(b) *The drying chamber and technical operating parameters:* Type of dryer and quantity, dryer capacity, time of use (week), chamber temperature (measure at 6 points), airflow (measure at 6 points), chamber dimension, building materials, energy source (LPG, electricity, biomass, diesel), electricity

　　consumption kWh/month, consumption of LPG (lb) and dollars/month, kg of biomass per dried, monthly diesel consumption (gallons), thermal problems, and maintenance date.

(c)　*Technical characteristics of the auxiliary equipment:* Brand, year of manufacture, quantity, power, and flow. The additional teams are extractors, resistors, motor, generator, and burner.

(d)　*Duct system, technical parameters, and power supply*: Dimension of the heat duct, inlet temperature in the duct system from the heat source, outlet temperature/connection to the chamber, inlet airflow, thermal problems, maintenance, and scheme of the dryer. Combustion chamber dimension (if applicable for biomass furnace), internal operating temperature, and heat exchanger outlet temperature. LPG burner dimension, number of nozzles, distance from the burner to the exhaust fan, connection to LPG cylinders, or reservoirs for industrial use. [30,31]

(e)　*Initial and final product temperature, humidity, and observations*: Product drying time, % initial and final humidity of the product (3 samples taken from each for analysis, the sample measured at the site), initial and final water activity of the product (3 samples measured at the site). Observations; environmental conditions [32]

　　It is necessary to indicate that 27 out of 68 programmed processing plants were valued, due to mistrust of the owners who prevented us from accessing their rice processing plants.

　　In the case of corn collection centres, 12 collection centres were valued The evaluation was carried out in the Los Ríos province's Quevedo sector where, only one dryer design (circular chamber dryer) was identified. Only the drying chamber area, depending on the amount of product to be dried, changes, while the energy source used in industrial LPG, the air extractor and the flame diffuser (nozzles) are of the same capacity for all dryers.

　　During the technical visit, it is shown that 60% of the equipment used in drying rice and 100% of the equipment used in drying corn do not have information plates. In some instances, burners and exhaust fans are locally built. Finally, during the evaluation, thermal losses were evident in the dryers and inadequate hot air distribution was evident in the drying chamber.

　　Figure 6 shows the mechanism that performs the measurements at the drying chamber (temperature meter with thermocouples type K), the area between the flame and the extractor (thermocouple type K-coated lance), and the product (humidity meter). Hot airflow, the chamber temperature, the duct system, and the dry product water activity were recorded during the technical visit. [30,31,33]. It should be noted that the equipment used for monitoring temperatures, wind speed, grain moisture meter, and water activity meter were previously calibrated.

　　Table 1 presents dryers' technical evaluation in the rice processing plants. It should be noted that there were several dryers in the second-generation plants; therefore, all the existing dryers in plants were assessed.

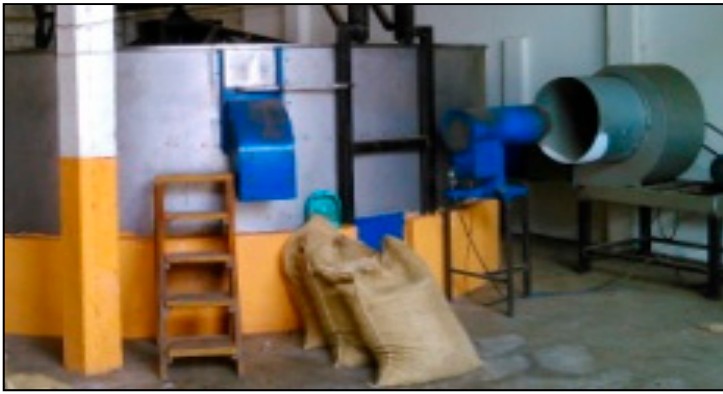

**Figure 5.** Parts of a grain dryer; drying chamber, burner, and exhaust fan.

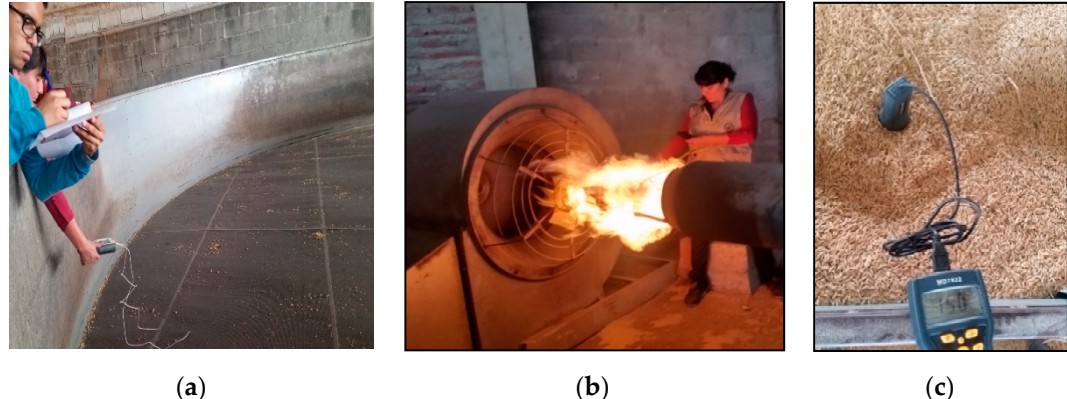

(**a**)            (**b**)            (**c**)

**Figure 6.** This figure shows (**a**) measurement of the drying chamber using a temperature recorder with type K thermocouples; (**b**) measurement of the inlet hot air temperature by the extractor, a ceramic-coated lance thermocouple; (**c**) measurement of the initial and final humidity of the product, spear type grain moisture meter.

**Table 1.** Assessment of the rice dryers.

| Dryer Evaluation | | | | | | |
|---|---|---|---|---|---|---|
| **Parameter** | **First Category (4 Dryers)** | **Dryers** | **Second Category (31 Dryers)** | **Dryers** | **Third Category (15 Dryers)** | **Dryers** |
| **Dryer type** | Elevator | 4 | Rectangular chamber<br>Rotary dryer for aged rice<br>Electric dryer for aged rice | 24<br>6<br>1 | Rectangular chamber | 15 |
| **Dryer capacity** | 13,636 kg | 1 | Chamber (LPG use):<br>2730–5452 kg | 22 | 724–908 kg | 10 |
| | 22,727 kg | 1 | Chamber (biomass use):<br>5454–10 909 kg | 2 | 908–1818 kg | 5 |
| | 40,909 kg | 1 | Rotary:<br>450 kg | 6 | | |
| | 54,545 kg | 1 | Electric:<br>1000 kg | 1 | | |
| **Drying time** | 12 h<br>12 a 14 h | 1<br>1 | 12–16 h<br>8 h (aged rice) | 24<br>7 | 12–15 h<br>14–24 h | 10<br>5 |
| **Chamber temperature** | 45–55 °C | | Chamber:<br>45–70 °C<br><br>Rotary:<br>100 °C<br><br>Electric:<br>100 °C | | 35–55 °C | |
| **Flame temperature** | Biomass use<br>300 °C | | LPG industrial: 280–600 °C<br><br>Biomass use:<br>250–380 °C | | Industrial LPG<br>250–350 °C<br>Domestic LPG<br>200–270 °C | |
| **Inlet temperature at the extractor** | | | 130–180 °C | | 80–170 °C | |
| **Chamber dimension** | 24 m$^2$<br>84 m$^2$ | 1<br>1 | 24–32 m$^2$<br>48 m$^2$ | 22<br>2 | 9–12 m$^2$<br>18–24 m$^2$ | 10<br>5 |
| **Energy source*** | Biomass<br>Diesel<br>Electricity | | Industrial LPG<br>Biomass<br>Electricity<br>Diesel (generator) | 28<br>2<br>31<br>1 | Industrial LPG<br>Domestic LPG<br>Electricity<br>Diesel (generator) | 1<br>14<br>12<br>3 |

Table 2 shows the results of the technical evaluation of corn dryers. Evaluation in 14 corn dryers (6 processing plants).

**Table 2.** Corn dryer assessment.

| Parameters | | #Dryers |
|---|---|---|
| Dryers Types | Circular chamber | 10 |
| | Rectangular chamber | 4 |
| Drying capacity | 13,670–22,730 kg | 10 |
| | 13,670 kg | 4 |
| Drying time | 6 h | |
| Chamber temperature | 44–80 °C | |
| Hot air temperature | Flame temperature 600 °C | |
| | Air duct temperature of 160–180 °C | |
| Chamber dimensions | 23–45 m$^3$ | 10 |
| | 22 m$^3$ | 4 |
| Air duct dimensions | 0.41–0.48 m$^2$ | |
| Wind speed | Extractor; 10–13 m/s | |
| | Chamber; 3–0.6 m/s | |

## 3. Results

The results of the technology evaluation the dryers used at the Ecuadorian national level for drying rice and yellow corn are presented.

### 3.1. Rice Processing Plant

### 3.1.1. Dryers Using Conventional Energies

The rice's drying is generally carried out in a square or rectangular chamber, made with masonry materials in its structural part (bricks and cement); trays use jute bags supported with square steel rods. The ducts and tubes for air intake are built with stainless steel plates. It should be noted that rotary type dryers and electric dryers also produce aged rice at operating temperatures of 100 to 120 °C, consuming LPG as main fuel (see Figure 7).

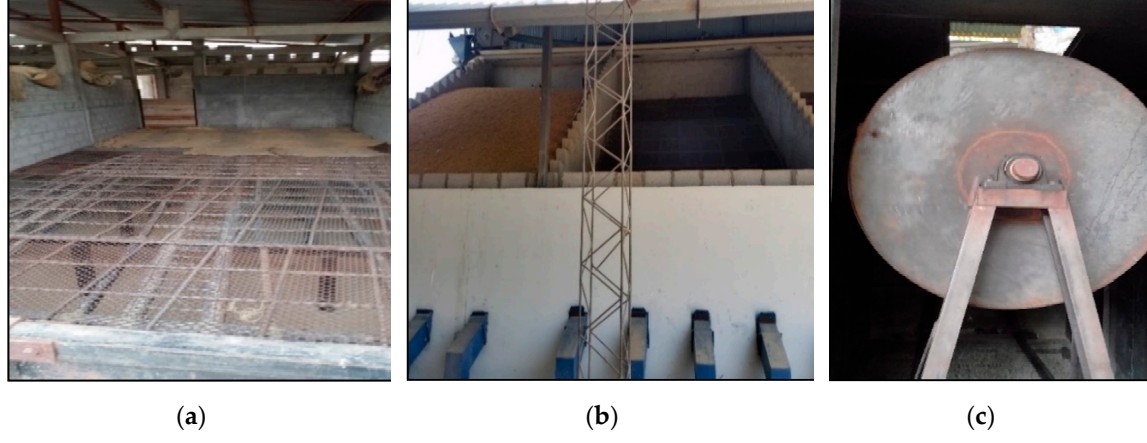

(**a**)  (**b**)  (**c**)

**Figure 7.** The figure shows (**a**) a dryer of a rectangular type chamber made with bricks and metal pipes; (**b**) a multichamber dryer made with perforated plates, bricks, and type C metal beams, (**c**) Gas rotary dryer, used for aging rice.

Another part of the dryer includes the heating system, consisting of a burner and an air extractor, built with local technology as shown in Figure 8.

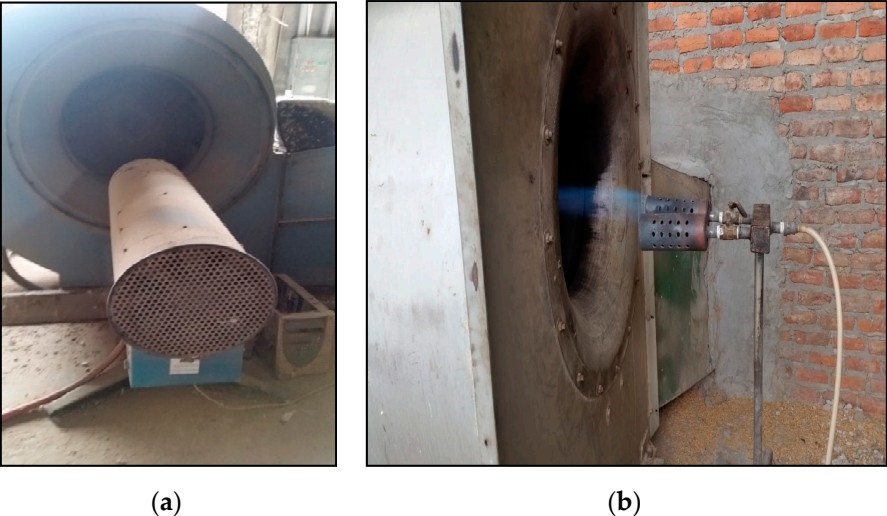

(**a**)          (**b**)

**Figure 8.** This figure shows the exhaust fan and types of gas burners handcrafted (**a**) 3-nozzle gas burner; (**b**) 2-nozzle gas burner.

In general, fossil fuel is commonly used to heat the air. The LPG is stored in industrial tanks (for larger capacity dryers) or household propane gas cylinders of 15 kg (small facilities for drying rice) (Figure 9).

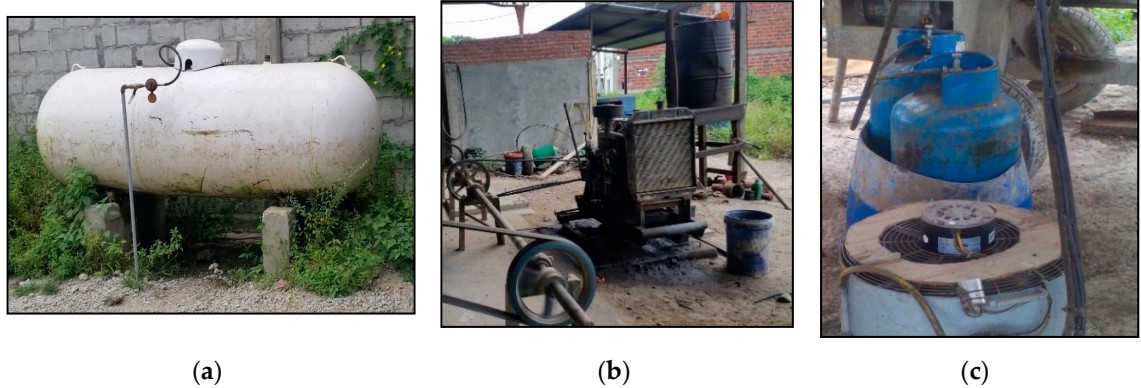

(**a**)          (**b**)          (**c**)

**Figure 9.** Types of fuels used as a heat source: (**a**) liquefied petroleum gas (LPG) tank for industrial use; (**b**) Diesel used for the operation of the generator; (**c**) LPG cylinder for domestic use.

### 3.1.2. Dryers Using Biomass

The first category processors currently use rice husk, as the primary fuel, and diesel ignition. Rice husk is the leading waste from the processing plant used to reduce fossil fuel costs in paddy rice drying process. Continuing along the same lines, very few second-class plants have also opted to use this type of biomass, building handcrafted combustion furnaces. Figure 10 shows imported industrial combustion furnaces and locally handcrafted furnaces.

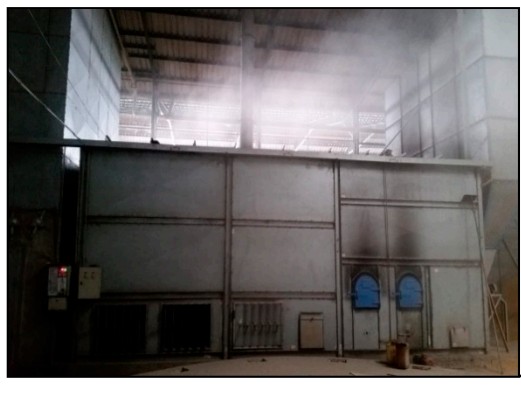
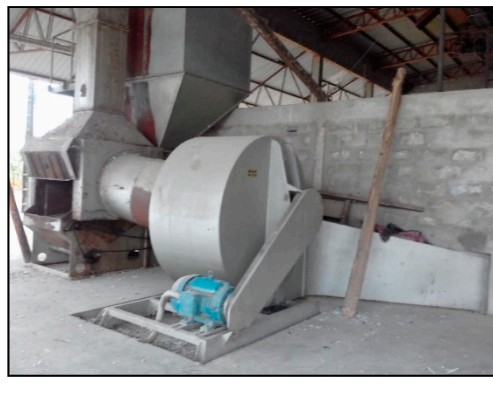

| (**a**) | (**b**) |

**Figure 10.** Shows two types of biomass dryers. (**a**) Double chamber dryer; on the left is the drying chamber and on the right is the combustion chamber. Brazilian technology, first-generation processing plant; (**b**) Handcrafted dryer with a drying chamber, combustion oven, ducts, and an exhaust fan in a second-generation processing plant.

There is a diversity of scientific and technological studies on rice husk about combustion, gasification, pyrolysis and chemical applications, as well as elaboration of products derived from this residue. In Latin America, the rice husk is use as fuel, construction, agriculture substrate, in the animal husbandry sector for bird beds and pigs, and due to its physical-chemical characteristics it is used for compost and fertilizers.

The calorific value of rice husk is similar to wood. Therefore, it serves for the manufacture of briquette or pellets. It is necessary to indicate that the use of the rice husk in a short-term combustion oven produces deterioration in the tubes of the heat exchanger and oven walls due to this biomass silica content.

The rice processing plant produces residue rice husk and rice dust quantified in percentages, that is to say, in a bag of rice with husk (equivalent to 200 pounds "90 kg"), 22% corresponds to mention residue. This rice husk is used primarily in the rice and sugar industry as a heat source for drying the raw material.

### 3.1.3. Energy Valuation

Table 3 presents the number of dryers evaluated in the first, second, and third categories of rice processing plants.

**Table 3.** Number of dryers evaluated in the three cantons of the Guayas province.

| Canton | Number of Dryers Evaluated | Category |
|---|---|---|
| Daule | 13 | (1) First, (5) Second, (7) Third |
| Salitre | 8 | (3) Second, (5) Third |
| Santa Lucia | 6 | (1) First, (5) Second |

Figure 11 presents a diagram of the energy consumption of a second and third category dryer.

Figure 11 shows that the surrounding air is heated by a gas burner (temperature of flame between 200 and 600 °C). The extractor removes the hot air from the environment (130 °C and 170 °C) to enter the drying chamber. Having performed a brief analysis, it is evident that there is an oversizing of the extractor and thermal loss in the square ducts that connect to the drying chamber.

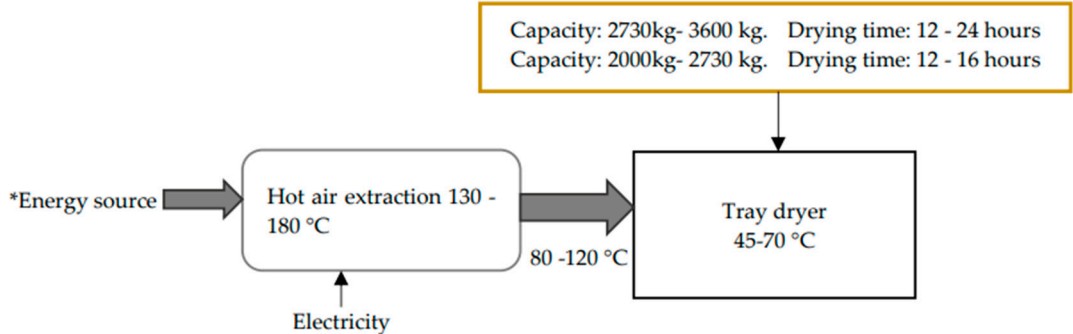

* Biomass
Hot gas temperature: 250 - 380 °C. 4 to 6 gallons of diesel required for flame ignition
* LPG
Flame temperature: 280 - 600 ° C

**a) rice processing plants of the second category**

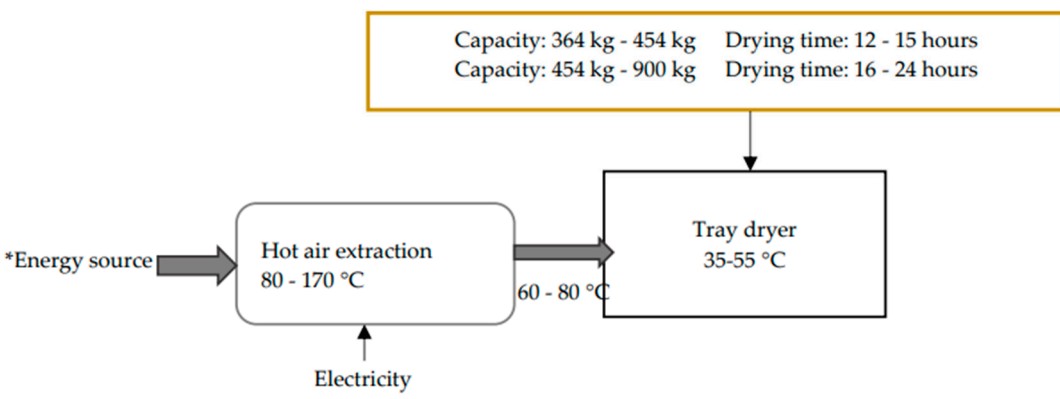

*LPG
Flame temperature: 150°-350°C

**b)  Rice processing plants of the third category**

**Figure 11.** Temperature and drying time (**a**) rice processing plants of the third category, (**b**) rice processing plants of the third category.

As an example, the rice dryer measurement results for a capacity of 909 kg are presented. Figure 12 shows that the chamber hot air temperature and speed are not uniform, causing the product to dry faster in one section than another. For more significant compression, in the rice dryer simulation, the observed flow is not turbulent in the entire chamber, due to the wind speed that initially carries at the entrance to the chamber (13 to 11 m/s). For this reason, it is necessary to place baffles in the post-chamber so that the hot air flow is distributing in the drying chamber.

Product drying operation takes from 12 to 24 h. The operation consists of turning on the equipment to dry the product for 2 to 3 h, then the burner is turned off, and the product is allowed to cool for one hour, then an operator moves the rice for 20 min. Finally, the burner starts again, and the cycle continues.

On the other hand, there are many setbacks in analysing energy consumption as the equipment does not have its corresponding information plates, this equipment being; extractors, motors, and fans. Therefore, the electricity consumption was estimated from their electrical bill, considering a 0.08 $/kWh reference cost. It should be noted that in rice processing plants, they do not only dry rice, they also peel the product. Therefore, the company energy cost is global. The actual energy consumption of drying and the power of the motors and extractors were estimated.

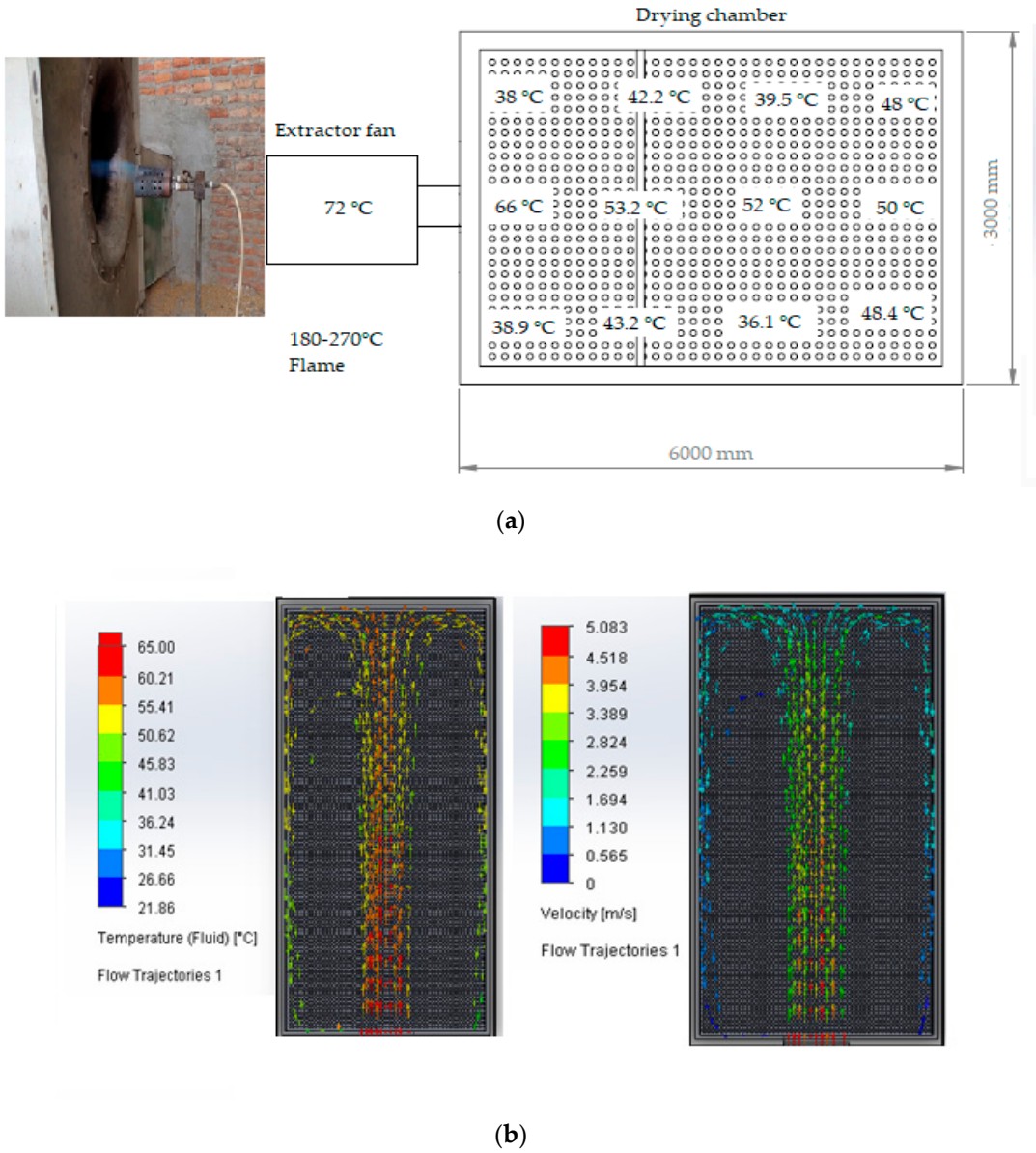

**Figure 12.** Drying chamber (**a**) dryer temperature measurements, (**b**) simulation of the hot airflow in the drying post-chamber.

For small drying centres, the LPG domestic tanks used for drying were considered. Since the domestic LPG is subsidized by the Ecuadorian government, its commercial price is $3 per LPG gas cylinder of 15 kg.

Concerning the dryers that use biomass as fuel, taking, for example, a dryer with a capacity between 1800 to 2726 kg of wet rice after 12 h of drying, it requires 1140 kg/h of husk and generates approximately 560 kg of ash. Finally, it is necessary to indicate that 90% of the dryers evaluated do not have a control system to set the hot air operating temperature entering the drying chamber; therefore, the dry product loss occurs around 2 to 4% by burning it.

Figure 13 shows a diagram of the biomass dryer of the first generation category.

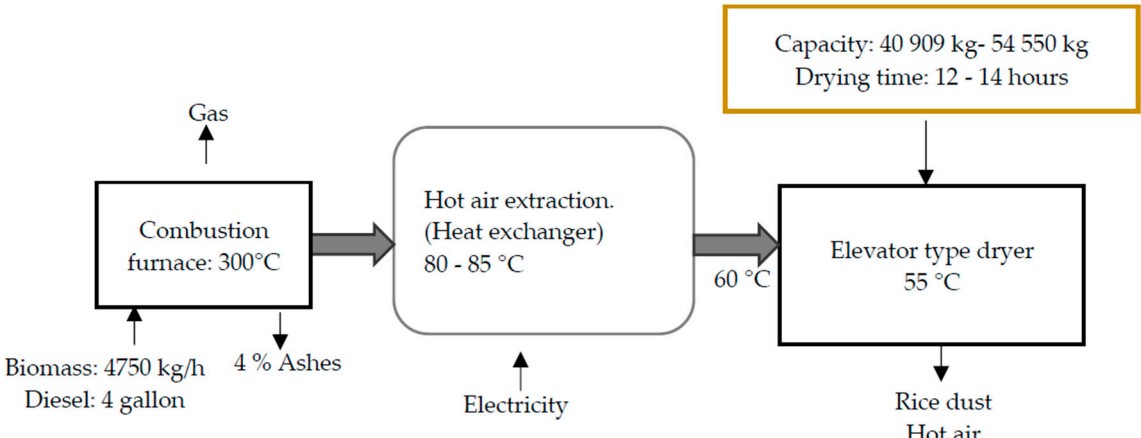

**Figure 13.** Temperature and dry time, using biomass. First category processing plant.

Table 4 shows the electricity, gas, biomass, and diesel consumption in rice processing plants.

**Table 4.** Energy consumption of the rice plant.

| Parameter | First Category 2 Processes | Quantity | Second Category 13 Processes | Quantity | Third Category 12 Processes | Quantity |
|---|---|---|---|---|---|---|
| Electricity consumption of the plant **/month | | | 1700–3500 kW/month | 4 | 1000–2000 kWh/month | 8 |
| | | | 3750–6000 kW/month | 9 | 2000–3500 kWh/month | 4 |
| | $900–$1500 | | $200–$360 | 4 | $40–$80 | 8 |
| | | | $400–$700 | 9 | $80–$150 | 4 |
| Gas consumption/dried | | | (kg) a tank without a consumption meter | 11 | 45–90 kg Domestic LPG | 11 |
| | | | | | 100–150 kg Industrial LPG | 1 |
| | | | $36–$100 | | $9–$18 Domestic LPG | 11 |
| | | | | | $21–$30 Industrial LPG | 1 |
| Biomass consumption*(rice husk)/dried | 378.8 kg/h–631.3 kg/h | 1 | 97–227 kg/h | 2 | | |
| | 1583.3 kg/h–2375 kg/h | 1 | | | | |
| Ash | 38 kg/h–56.8 kg/h | 1 | 9.7–24 kg/h | | | |
| | 110 kg/h–166.3 kg/h | 1 | | | | |
| Consumption * Diesel fuel/dried | | | biomass dyers | | | 3 |
| | 2–5 gal. | | 3–6 gal diesel | 2 | Generator 20–38 gal | |
| | | | Generator | | | |
| | | | 25–30 gal | 1 | | |
| | $3–$7.5 | | Biomass dryers $4.5–$9 | | Generator $21–$40 | 3 |
| | | | Generator $26–$31 | | | |

\* It is counted by the number of rice processing plants. ** Total electrical consumption of the rice processing plants.

Analysing Table 4, for first category rice plants when using rice husk as heat source, the main source of consumption is the electricity used to operate the extractor motors (1 to 5 HP) and bands. Moreover, the Table 4 analyses the amount of biomass dryer energy consumption, as a production indicator per equipment operating cycle. Table 5 shows the estimated production indicator; for a better understanding of the analysis, it has been considered to present the categories by letters; letter A for dryers in first category plants (elevator dryers), letter B for dryers in second category plants,

and letter C for third category plants. The information is presented in quintals (45.45 kg) which is the measurement that the product is commercialized.

**Table 5.** Production indicator.

| Dryer | Capacity (Quintal) | Drying Hours | Electricity Consumption per Month (Dollars) | LPG per Month (Dollars) | Diesel per Month (Dollars) | Rice Husk kg/h | Ash (kg) kg/h | Production Indicator (Dollars/Quintal) |
|-------|-------|-------|-------|-------|-------|-------|-------|-------|
| A1 | 600 | 12 | 450 | —— | 60 | 568 | 39.8 | 0.59 |
| A2 | 2400 | 12 | 750 | —— | 100 | 1818.2 | 127 | 1.4 |
| B1 | 100 | 14 | 80 | 200 | — | — | — | 0.18 |
| B2 | 240 | 14 | 339 | —– | 99 | 155.4 | | 0.27 |
| B3 | 500 | 12 | —— | 1500 | 250 | — | — | 0.14 |
| C1 | 20 | 16 | 20 | 192 | —— | —— | — | 0.05 |
| C2 | 20 | 30 | —· | 144 | 51 | —— | — | 0.05 |

As shown in Table 5, the production indicator for drying a rice is more efficient in higher capacity biomass dryers; it is noticeable that small dryers using domestic LPG have a very low production indicator; on the other hand, the drying time can take 30 h for drying the rice, starting from the initial humidity of 21–28% to reach 10–11% of final humidity.

For this analysis, two dryers have been selected. The first uses a biomass heat source, and the second LPG has capacities of 240 quintals (10,909.0 kg) and 500 quintals (22,723.3 kg), respectively. The calculation starts by defining the water total evaporation rate in the product $V_T$ (kg$_{water}$/h) [1] [34,35]

$$V_T = W_0 * (X_{wb1} - X_{wb2})/(1 - {}_{Xwb2})/t \qquad (1)$$

where: Wo is the initial weight of solid wet in kg; $X_{wb1}$ is the initial product moisture; $X_{wb2}$ final product moisture and t drying time in hours.

Performance is determined by the relationship between the minimum kg required to evaporate water from a given mass of grains and the amount of kcal consumed [36].

$$Performance = (V_T * C_L)/(M_{fuel} * LCV) \qquad (2)$$

Being, $C_L$; latent heat of vaporization of the water 540 kcal/kg. $M_{fuel}$; fuel mass use in kg/h. LCV; lower calorific value in kcal/kg, being 11,082 kcal/kg for LPG [35,36], rice husk 3577 kcal/kg–kJ/kg [37].

Table 6 determines that the exergetic performance of the dryer using LPG is superior by directly heating the air entering the drying chamber (Figure 11); in relation to using a biomass source, a combustion furnace needs to burn the shell, heat is transferred to an exchanger where the air entering the dryer is heated (Figure 13). Therefore, there is loss of performance from the combustion chamber, heat exchanger, and ducts.

**Table 6.** Exergetic performance of the dryers.

| Dryer | Capacity (Quintal) | $W_0$ (kg) | t (h) | $X_{wb1}$ (%) | $X_{wb2}$(%) | $C_L$ kcal/kg | $M_{fuel}$ kg/h | LCV kcal/kg | Performance% |
|-------|-------|-------|-------|-------|-------|-------|-------|-------|-------|
| Biomass | 240 | 10,909 | 14 | 23 | 11 | 540 | 155.4 | 3577 | 10 |
| GLP | 500 | 22,727.3 | 12 | 21 | 11 | 540 | 32.2 | 11,082 | 32.2 |
| Biomass | 600 | 27,273.7 | 12 | 21 | 11 | 540 | 567 | 3577 | 6.8 |

## 3.2. Corn Processing Plants

Corn is one of the most important cereal crops in Ecuador and the world due to its high productivity, chemical composition, and nutritional value. On the other hand, it is considered that the waste generated during harvest and post-harvest could be used as a source of thermal energy. Among the main causes as to why this biomass source has not been used until now are different technical difficulties and lack of sufficient information on the waste quantity and processing [38,39].

Borja Velázquez indicates in the book "Exploitation of Biomass for Energy Use" that dispersed biomass makes obtaining the final product more expensive; for this reason, collection, transport, and storage should be considered [40].

Yellow corn is usually dried naturally (sun-dried) or using artificial dryers. The difference between both processes is the grain brightness and drying time. The product has a brighter and yellow appearance when naturally dried; on the other hand, artificial drying gives an opaque appearance and colour to the product, which affects its sale.

The corn drying technology is the same for all corn collection centres, consisting of; a circular/rectangular drying chamber, a gas burner, and an extractor. It is varying the dimensions of the chamber of this equipment from the capacity of the dryer. Figure 14 shows the drying technology used in the sector, and Figure 15 shows the temperature values recorded in the drying chamber as an example. As you can see, the temperature is not uniform in the chamber, varying between 32 °C and 76 °C in certain parts.

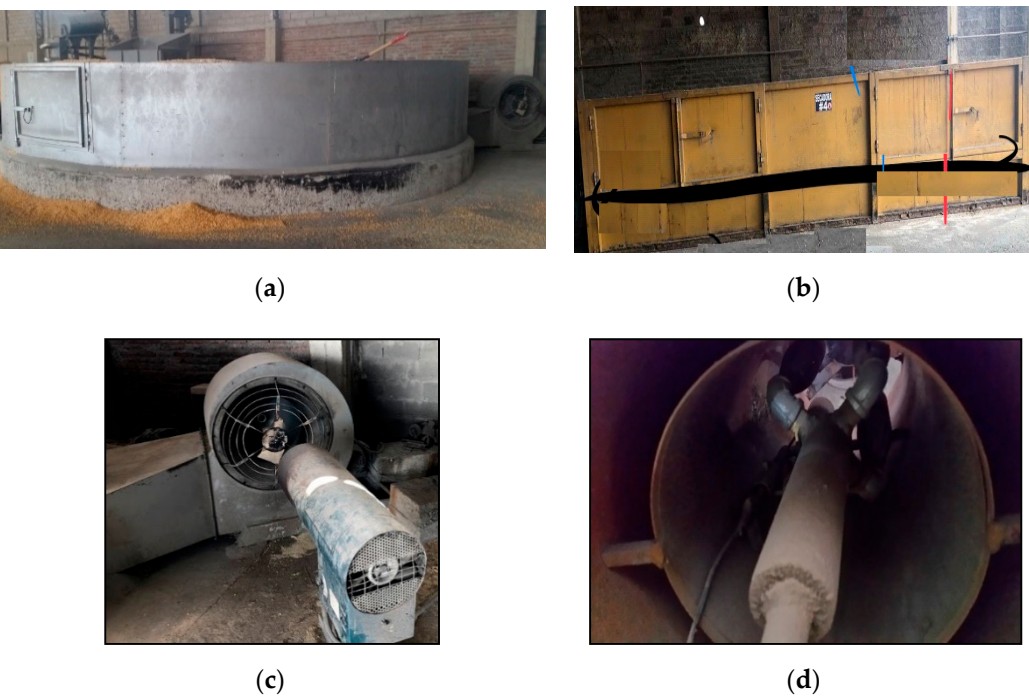

**Figure 14.** Corn dryers: (**a**) circular chamber; (**b**) square camera; (**c**) burner and exhaust fan; (**d**) four nozzle burner.

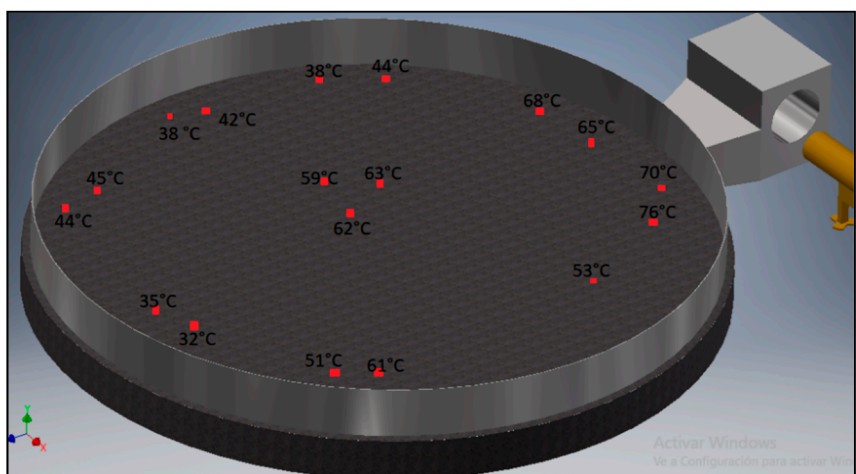

**Figure 15.** Chamber temperature. The capacity of 400 quintals (18,200 kg).

Table 7 shows the results of the energy consumption assessment in 14 corn dryers (6 processing plants).

**Table 7.** Energy consumption in the corn dryer.

| Parameters | Results |
|---|---|
| Energy source | Industrial LPG Electricity |
| The electric consumption of the plant/monthly | $450–$800 |
| Consumption of gas of the plant/monthly | $3000–$4000 |

On Table 7, the corn dryers of the medium production collection centres are evaluated. Small producers sun dry yellow corn; the use of technology is not profitable due to the fuel and electricity consumption required by current drying equipment. Compared to the rice processing plants' case, this sector also has the same problem in the extractor underdimensioning, consuming more than what is required in the electrical part, consuming around 2000 to 4000 kWh/month. This added that the dryers of corn spend between 3000 to 4000 dollars/month to dry between 300 to 500 quintal (13,670 kg to 22,730 kg), representing 20 dries per month.

Analysing the production indicator in yellow corn dryers with the same drying capacity as shown in Table 8, the production indicator is 0.27 to 0.47 dollars/quintal, showing that the dryer engine and LPG burner of the A2 dryer are oversized.

**Table 8.** Production indicator (drying corn).

| Dryer | Capacity (Quintal) | Drying Time (Hours) | Drying per Month | Electricity Consumption per Month (Dollars) | Industrial LPG per Month (Dollars) | Production Indicator Dollars/Quintal |
|---|---|---|---|---|---|---|
| A1 | 400 | 6 | 3 | 100 | 760 | 0.47 |
| A2 | 400 | 6 | | 286.5 | 1200 | 0.27 |

Entering the technical context, evaluating the flame front temperature is around 400 to 600 °C and hot air is drawn into a duct at temperatures reaching 180 °C, into the chamber. During this stage, there are thermal losses in the duct, added to the operator insecurity due to the burner flame as it does not have a flame protection cover.

The data in Table 7 proceeds to determine two dryers exergetic performance with the same wet product drying capacity. Using Equation (2), previously expressed CL; latent heat of vaporization of the water 600 kcal/kg [35], LCV; lower calorific value of LPG in kcal/kg, 11,082 kcal/kg [38].

Table 9 presents the calculation analysis and the results. The performance of the dryer A2 was 25%, which leads to energy losses of 75% focused on the flame environment when heating the outdoor air on extractor sizing and uninsulated ducts.

**Table 9.** Exergetic performance of the corn dryer.

| Dryer | Capacity (Quintals) | $W_0$ (kg) | t (h) | $X_{wb1}$ (%) | $X_{wb2}$ (%) | $C_L$ kcal/kg | $M_{fuel}$ kg/h | LCV kcal/kg | Performance % |
|---|---|---|---|---|---|---|---|---|---|
| A1 | 400 | 18,181.8 | 6 | 19.4 | 13 | 600 | 28 | 11,082 | 43.1 |
| A2 | 400 | 18,181.8 | 6 | 20 | 13 | 600 | 53 | 11,082 | 25 |

## 4. Discussion

This study objective was to establish the diagnosis concerning the energy consumption used in current drying in small and medium-scale plants. From a technological point of view, the drying process presents options or variables that allow optimization consuming energy while maintaining the same quality of the final products.

The evaluation results of the energy consumption in the corn and rice dryers shows that an energy efficiency savings plan is required, starting from the redesign of the LPG consuming dryers based on the actual capacity of the equipment and also installing a thermostat and humidity regulator to control the drying time.

The rice small-scale dryers consume domestic LPG as an energy source, considered to be a problem of interest to the Ecuadorian State because fuel is subsidized.

Considering the variable costs of the drying process which are electricity and LPG fuel costs, it was obtained that the variable cost of drying is 1.1 a 1.9 dollars/quintal (45.45 kg). It is worth mentioning that, to obtain an integral cost of drying, information on fixed expenses such as permanent labour, the value of the drying equipment, and useful lifetime is needed [33,41].

Very little information has been published at the bibliographic level regarding energy conservation potential in industrial drying. The importance of sustainable energy use forces the rethinking of traditional approaches where the dryer is only using LPG as energy source.

In practice, most drying industries ignore energy efficiency, arguing that product quality and performance are paramount. However, an appropriate balance must be maintained between these parameters. The process of designing and improving the dryer must be continuously reviewed. Many external factors can change in the dryer's useful lifetime and therefore affect the economics of the whole company.

For example, most industries consume about 25% of the total energy used in product processing, where the cost of drying can approach 60–70% of the total cost [42,43]. Energy demand for drying processes and, in particular, for product processing increases each year as production increases [44]. This increase in energy obliges the search for technological solutions that optimize conventional energy use and that allow the transition to hybrid generation systems based on renewable energies.

As discussed in the current literature, the cost-benefit analysis concerning improvements from the use of biomass as an energy source has proved to be very interesting in short recovery periods with significant savings [42,45–47]. However, its application is not very common in companies with small rice production volumes and is almost incipient in corn. At the regional level, several authors cite that the drying industry should improve control at the plant to achieve modernization, increase exegetic efficiency [48], and reduce reliance on conventional energy vectors.

The demand of energy for rice and corn processing is increasing in Latin America. Therefore, to achieve sustainable energy management, rice and corn drying industries must search for mechanisms and alternatives to reduce energy consumption, increasing process efficiency. This would lead to a reduction in emissions and an increase in biomass supply as an energy source.

It is important to mention that an exhaustive analysis of the available information reveals inconsistencies from the terminology of the process, technological aspects, and the interpretation of variables and parameters. The reported data on technology varies significantly and frequently contradict both drying theory and industrial practices.

The percentage values of the calculation of exergy considering renewable and conventional energy sources determined in this study are within the reference ranges of other authors [49,50]. They show that there are essential exergy losses generated in evaporation, ambient (surroundings), airflow, radiation, and convection losses in the dryers. The results indicated that about 10 to 32% of exergy is used for drying rice and 25 to 43% for drying corn, losses waste the remaining energy. It is necessary to establish actions to increase the process exergy, considering a renewable energy source, feed airflow management, and combustion chamber design.

Finally, improper energy management in drying processes can influence the products supply chain, causing fluctuations in its final price. Consequently, the industries where products such as rice and corn are processed must establish future energy efficiency actions to improve their operations, especially drying, thus ensuring their competitiveness in the markets, both local and international.

## 5. Conclusions

The drying technology for the products considered in our work evidenced the inadequate dimensioning of the hot air extractors and burner, especially for corn drying, which leads to high electricity and LPG consumption. Therefore, it is the industrial sector where there is the most energy waste. On the other hand, first-class rice processing plants have chosen to change their technology using biomass sources, followed to a lesser extent by second-class plants. Among the observations related to biomass combustion, the inadequate handling of the ashes by the operators increases the risk of causing long-term health problems.

Regarding the drying temperature of the products, from the literature, the drying of rice and corn would be around 50 °C, so that they do not lose their organoleptic properties. The research evidenced that the drying is carried out at higher temperatures than those suggested in the bibliography.

Finally, this article provides an overview of the different existing drying technologies as well as energy management. Once the drying technology has been analysed, it can be concluded that it is essential to start with technological innovation for the agroindustrial sector based on energy efficiency for saving fuel and electricity in drying equipment.

**Author Contributions:** Conceptualization, B.V.-M.; methodology, E.D.-P., J.P.-J., and M.Q.; validation, E.D.-P., and J.P.-J.; formal analysis, J.P.-J.; investigation, M.Q.; writing—original draft preparation, E.D.-P.; validation and writing—review and editing, H.A. All authors have read and agreed to the published version of the manuscript.

**Funding:** This research was funded by Red Ecuatoriana de Universidades y Escuelas Politécnicas para el desarrollo de la Investigación y Posgrados.

**Acknowledgments:** Technical support from a las Red IBEROMASA and Red Iberoamericana de Eficiencia termica industrial (RIETI).

**Conflicts of Interest:** The authors declare no conflict of interest.

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
