# Peer review of "Estimation of the Energy Consumption of the Rice and Corn Drying Process in the Equatorial Zone"

_applsci, doi:10.3390/app10217497_

Round 1
Reviewer 1 Report
The manuscript is a document aimed at the agricultural sector of Ecuador that can draw attention to unnecessary excessive energy use and problems with rice and maize drying equipment to help farmers improve this. Research is very important; they come down to finding specific problems in specific groups of producers. The authors described the climatic influence, wind; I propose to take into account the humidity and temperature of the air in the surroundings and inside the dryer in subsequent tests, also depending on the time of day. However there is no novelty in the field of drying or new solutions, e.g. based on solar energy with collectors.
Collecting the data was not easy, because especially small producers of dried grains have little knowledge of drying, energy, new devices, or do not want to provide such data. The purposefulness of the conducted research was proved and justified.
The manuscript is valuable but needs improvement in many places, e,g. units, a series of numerical errors related to confusing "dot" with "comma”. Many errors in writing values and units make the results unreliable.
"0.27 quintal/dollar for corn, using biomass reaches 1.4 sacks/dollar" - somewhat unusual indicators, better the opposite - dollars/sack ...
In all tables, units should only follow the pointer names instead of each value
There should be the same unit of the same indexes throughout the manuscript, see Lines 25, 84, 105..
In my opinion, the manuscript is too extensive, the Authors should shorten it, especially the description of the regional specificity of Ecuador (in the Introduction section), but it should also be organized as a whole. Some of the data is in "Methods" and some in "Result".
Line 38: “..the energy requirement for drying food grains has two components: first is the energy required to evaporate the free water and second is the energy required to remove the water associated with the raw material being processed..” - Rewrite this sentence; it is not possible to remove all the bound water. I propose to mention two periods of drying (drying kinetics) of food here.
Line 58: “..population of 17,096,789 58 million inhabitants..” - rewrite that number or unit
Chapter "Methods" - it is difficult to assess which methods it applies to. At the beginning of this chapter, it is worth defining a research plan. Authors should present a scheme to specify the next stages of the research. Everything is written descriptively, it should be impersonal, and there are phrases such as "we proceed to investigate", “For this study, we will address ourselves,”, “Continuing with the analysis, we proceed to carry out a review of..”.
Line 514: "latent heat of vaporization of the water 600 kcal / kg [37]" ..?- There is no such information in this source. In line 453 and Tab. 5 there is the value "latent heat of vaporization of the water 540kcal/kg"? - was there a different pressure?
Conclusions is a further part of the Discussion..
References consist of 51 items, but they are written carelessly, missing titles, page numbers, capital letters (line 644), lower case letters in the journal name (line 614), month (lineb 696), sometimes there are quotation marks, missing or excess spaces, also dots and others. Sometimes the authors have their full names, but most have initials, as is the case for journal abbreviations..

Author Response
- The manuscript is valuable but needs improvement in many places, e,g. units, a series of numerical errors related to confusing "dot" with "comma”. Many errors in writing values and units make the results unreliable.
Modified text
- "0.27 quintal/dollar for corn, using biomass reaches 1.4 sacks/dollar" - somewhat unusual indicators, better the opposite - dollars/sack ...
The modifications dollars / quintal and dollars / kg ,have been made
In all tables, units should only follow the pointer names instead of each value
- There should be the same unit of the same indexes throughout the manuscript, see Lines 25, 84, 105..In my opinion, the manuscript is too extensive, the Authors should shorten it, especially the description of the regional specificity of Ecuador (in the Introductionsection), but it should also be organized as a whole. Some of the data is in "Methods" and some in "Result".Line 38: “..the energy requirement for drying food grains has two components: first is the energy required to evaporate the free water and second is the energy required to remove the water associated with the raw material being processed..” - Rewrite this sentence; it is not possible to remove all the bound water. I propose to mention two periods of drying (drying kinetics) of food here.
Modified
- Line 58: “..population of 17,096,789 58 million inhabitants..” - rewrite that number or unit
Modified
- Chapter "Methods" - it is difficult to assess which methods it applies to. At the beginning of this chapter, it is worth defining a research plan. Authors should present a scheme to specify the next stages of the research. Everything is written descriptively, it should be impersonal, and there are phrases such as "we proceed to investigate", “For this study, we will address ourselves,”, “Continuing with the analysis, we proceed to carry out a review of..”.
Modified
- Line 514: "latent heat of vaporization of the water 600 kcal / kg [37]" ..?- There is no such information in this source. In line 453 and Tab. 5 there is the value "latent heat of vaporization of the water 540kcal/kg"? - was there a different pressure?
From the FAO bibliography, it establishes that the latent heat of vaporization of the water for the corn is 600 kcal / kg
Conclusions is a further part of the Discussion..
Modified
References consist of 51 items, but they are written carelessly, missing titles, page numbers, capital letters (line 644), lower case letters in the journal name (line 614), month (lineb 696), sometimes there are quotation marks, missing or excess spaces, also dots and others. Sometimes the authors have their full names, but most have initials, as is the case for journal abbreviations..
Modified

Reviewer 2 Report
The authors presented the results of the evaluation of the energy consumption of rice and maize dryers in Ecuador. The manuscript appears to be original. The topic is interesting for readers as well as for the industrial sector because it is utilitarian.
However, the manuscript requires significant improvement to make it understandable in terms of the methods and results presented. The introduction gives a short overview information, which are needed to introduces the research question. However, the information on methods and types of dryers used for drying rice and maize should be supplemented in this section. At the end of the Introduction, please add information about the purpose of the research.
The methodological part is very poor and should be revised to include all the information necessary to understand the design of the experiment, which seems to be appropriate for examining the research issue in terms of repetition, sample independence and data handling. Some queries presented below need to be answered by authors:
Paragraphs 2.1.1 and 2.1.2 describe in detail the first, second and third categories of processing plants in terms of the methods used and the types of dryers. The lack of this information makes it difficult to compare the assessed dryers and to draw conclusions about the most energy-intensive rice and maize dryers. Part of the table No. 2 - Dryer evaluation should be moved to the section of the method. The same should be done with the data on the assessed maize dryers, the information should also be included in this section.
In tables 3 and 4 (Results part) the abbreviations A1, A2 and B1, B2… .. and C1… .. should be explained. Dryers with such symbols have not been characterized before. This should be done in the methodological part.
Line 24 and 25: The results should be presented in terms of weight of rice and maize (kg), not sack.
Line 24: expand the abbreviation LPG
Line 24: the unit 0.27 quintal / dollar should be written as 0.27 quintal dollar-1. Please analogously replace the entries of units throughout the article.
Line 30: Corn needs to be added.
Line 438: Results should be presented in terms of weight of rice and corn (kg), not sack.

Author Response
- Paragraphs 2.1.1 and 2.1.2 describe in detail the first, second and third categories of processing plants in terms of the methods used and the types of dryers. The lack of this information makes it difficult to compare the assessed dryers and to draw conclusions about the most energy-intensive rice and maize dryers. Part of the table No. 2 - Dryer evaluation should be moved to the section of the method. The same should be done with the data on the assessed maize dryers, the information should also be included in this section.
Modified
- In tables 3 and 4 (Results part) the abbreviations A1, A2 and B1, B2… .. and C1… .. should be explained. Dryers with such symbols have not been characterized before. This should be done in the methodological part.
Table 5 shows the estimated production indicator; for a better understanding of the analysis, it has been considered to present the categories by letters; letter A for dryers in first category plants (elevator dryers), letter B for dryers in second category plants, and letter C for third category plants.
- Line 24 and 25: The results should be presented in terms of weight of rice and maize (kg), not sack.
Modified
- Line 24: expand the abbreviation LPG
Modified
- Line 24: the unit 0.27 quintal / dollar should be written as 0.27 quintal dollar-1. Please analogously replace the entries of units throughout the article.
Modified
- Line 30: Corn needs to be added.
ok
- Line 438: Results should be presented in terms of weight of rice and corn (kg), not sack.
Modified in quintal

Reviewer 3 Report
The paper: “Estimation of the energy consumption of the rice and corn drying
process in the equatorial zone” by Emérita Delgado-Plaza , Miguel Quilambaqui, JuanPeralta-Jaramillo, Héctor Apolo, Borja Velázquez-Martí present data about the comparison of the energy consumption of different rice and maize dryers. The subject is interesting but the paper needs to be deeply improved before publication.
Some general suggestions:
- An extensive English revision is needed to improve the readability of the text.
- There are too many figures (19): I suggest to choose the most relevant and to group them in panels.
Fig 3 and Fig 4: the data can be added to the text and these figures can be deleted.
Fig 6: Fig6 a, b and c should have the same dimension, the same data are presented in table and histogram, choose table or histogram. If “Quantity” is the total, the number isn’t correct in column “first category” of fig 6a,b,c.
Fig 14 and 15 are difficult to interpret.
Table 2 should be set up as Table 3,4,5 etc.. and split in Table 2A (Assessment of rice dryers) and Table 2B (Energy consumption).
Table 6 should be set up as Table table 3,4,5 etc.
- 24 references out 51 are not in English.
- Introduction
The text should be revised by an English native to improve the readability:
As example:
Line 36-38: the energy requirement for drying food grains has two components: the energy required to evaporate the water content in the grains and the energy required to remove the water associated with the raw material [2][3].:
Line 40-46:
In this sense we can define it as a process, natural or artificial, where there is a synchronous exchange of heat and massbetween the air present in the drying process and the grains. In other words, we can call drying the
process capable of reducing the moisture content of a product to a minimum and safe level so that it can be consumed and stored. In this way, the respiratory activity of the grain is reduced, preventing the attack of insects fungi and molds, that, reducing the quality of the grain, prevents its commercialization and consumption [2][3][5].
- Methods section should be deeply revised: methods should be presented in a more schematic way, background data, results and comments moved to Introduction/Results/Discussion sections.
- Results section need a deep revision: the data should be presented in a more concise and clear form. Furthermore I suggest to move the section 3.1.1 and the first part of section 3.2 (lines 464-475) to Introduction.
- As for the previous sections also Discussion needs not only an English revision but also general revision of the data discussed. When data are commented, the relative Figure/Table should be cited. I suggest to move the first part of Conclusion (line 573-581) to top of Discussion, Conclusions should be more concise.

Author Response
- An extensive English revision is needed to improve the readability of the text.
Text modifications have been made
- There are too many figures (19): I suggest to choose the most relevant and to group them in panels.
The number of figures has been reduced (15 figure)
3-Fig 3 and Fig 4: the data can be added to the text and these figures can be deleted.
Figures have joined
4- Fig 6: Fig6 a, b and c should have the same dimension, the same data are presented in table and histogram, choose table or histogram. If “Quantity” is the total, the number isn’t correct in column “first category” of fig 6a,b,c.
The histogram data represent the number of center in the province of Guayas, Figure 4 (above 6) shows the number of dryers in the areas of Daule, Salitre and Santa Lucia.
5- Fig 14 and 15 are difficult to interpret.
modified
6- Table 2 should be set up as Table 3,4,5 etc.. and split in Table 2A (Assessment of rice dryers) and Table 2B (Energy consumption).
The table has been divided into, table 1 and table 4
7- Table 6 should be set up as Table table 3,4,5 etc
The table has been divided into, table 2 and table 7
8-. 24 references out 51 are not in English
References are presented in Spanish
- Introduction
The text should be revised by an English native to improve the readability:
As example:
Line 36-38: the energy requirement for drying food grains has two components: the energy required to evaporate the water content in the grains and the energy required to remove the water associated with the raw material [2][3].:
modified
Line 40-46:
In this sense we can define it as a process, natural or artificial, where there is a synchronous exchange of heat and massbetween the air present in the drying process and the grains. In other words, we can call drying the
process capable of reducing the moisture content of a product to a minimum and safe level so that it can be consumed and stored. In this way, the respiratory activity of the grain is reduced, preventing the attack of insects fungi and molds, that, reducing the quality of the grain, prevents its commercialization and consumption [2][3][5].
modified
- Methods section should be deeply revised: methods should be presented in a more schematic way, background data, results and comments moved to Introduction/Results/Discussion sections.
Part of the methods has been moved to the introduction
11- Results section need a deep revision: the data should be presented in a more concise and clear form. Furthermore I suggest to move the section 3.1.1 and the first part of section 3.2 (lines 464-475) to Introduction.
The methodology has been changed to follow a logical consequence in the results
12. As for the previous sections also Discussion needs not only an English revision but also general revision of the data discussed. When data are commented, the relative Figure/Table should be cited. I suggest to move the first part of Conclusion (line 573-581) to top of Discussion, Conclusions should be more concise.
modified

Round 2
Reviewer 2 Report
The manuscript now seems much better than in its first version although significant revision is still necessary:
- -Lines 23-24: the unit 0.27 quintal / dollar should be written as 0.27 quintal
dollar-1. Please analogously replace the entries of units throughout the article. - Line 36, 43…- references in the text should be written as [2, 3], not [2], [3]. Please analogously replace the unit entries throughout the article.
Author Response
1- -Lines 23-24: the unit 0.27 quintal / dollar should be written as 0.27 quintal dollar-1. Please analogously replace the entries of units throughout the article
Corrections have been made in dollars/quintal for a better understanding
2- references in the text should be written as [2, 3], not [2], [3].
Corrections have been made
Reviewer 3 Report
The paper: “Estimation of the energy consumption of the rice and corn drying process in the equatorial zone” by Emérita Delgado-Plaza, Miguel Quilambaqui, JuanPeralta-Jaramillo, Héctor Apolo, Borja Velázquez-Martí, has been revised but in my opinion it’s not yet suitable for publication, in particular an extensive English revision is still needed. Too many sentences throughout the paper are not written clearly:
For example Line 17-22:
Being a topic of interest to the agricultural sector, it (?) has evaluated rice and maize dryers' energy consumption in the area (?). To meet this goal, developed (?) an overview survey matrix and protocols for temperature and wind speed measurements of dryers were developed; the study evaluated 49 rice dryers and 14 yellow corn dryers. As a result (?) , the oversizing of the fan/extractor and engine of the dryer generates a high energy consumption, added to the lack of insulation in the heat ducts.
Author Response
Text corrections have been made